# A Review of the Recent Advances in Alzheimer’s Disease Research and the Utilization of Network Biology Approaches for Prioritizing Diagnostics and Therapeutics

**DOI:** 10.3390/diagnostics12122975

**Published:** 2022-11-28

**Authors:** Rima Hajjo, Dima A. Sabbah, Osama H. Abusara, Abdel Qader Al Bawab

**Affiliations:** 1Department of Pharmacy, Faculty of Pharmacy, Al-Zaytoonah University of Jordan, P.O. Box 130, Amman 11733, Jordan; 2Laboratory for Molecular Modeling, Division of Chemical Biology and Medicinal Chemistry, Eshelman School of Pharmacy, The University of North Carlina at Chapel Hill, Chapel Hill, NC 27599, USA; 3National Center for Epidemics and Communicable Disease Control, Amman 11118, Jordan

**Keywords:** Alzheimer’s disease (AD), diagnostic biomarkers, drug prioritization, epigenetics, network biology, multi-target drug ligands (MTDLs), disease pathways

## Abstract

Alzheimer’s disease (AD) is a polygenic multifactorial neurodegenerative disease that, after decades of research and development, is still without a cure. There are some symptomatic treatments to manage the psychological symptoms but none of these drugs can halt disease progression. Additionally, over the last few years, many anti-AD drugs failed in late stages of clinical trials and many hypotheses surfaced to explain these failures, including the lack of clear understanding of disease pathways and processes. Recently, different epigenetic factors have been implicated in AD pathogenesis; thus, they could serve as promising AD diagnostic biomarkers. Additionally, network biology approaches have been suggested as effective tools to study AD on the systems level and discover multi-target-directed ligands as novel treatments for AD. Herein, we provide a comprehensive review on Alzheimer’s disease pathophysiology to provide a better understanding of disease pathogenesis hypotheses and decipher the role of genetic and epigenetic factors in disease development and progression. We also provide an overview of disease biomarkers and drug targets and suggest network biology approaches as new tools for identifying novel biomarkers and drugs. We also posit that the application of machine learning and artificial intelligence to mining Alzheimer’s disease multi-omics data will facilitate drug and biomarker discovery efforts and lead to effective individualized anti-Alzheimer treatments.

## 1. Introduction

Alzheimer’s disease (AD) is a polygenic and multifactorial disease characterized by the deposition of amyloid-β (Aβ) fibrils in the brain, leading to the formation of plaques and neurofibrillary tangles (NFTs), and ultimately resulting in dendritic dysfunction, neuronal cell death, memory loss, behavioral changes, and organ shutdown [1,2,3,4,5]. It is estimated that 6.5 million Americans, 65 years and older, are living with Alzheimer’s dementia, and this number is likely to grow rapidly [6], reaching 13.8 million by 2060 [6,7]. Additionally, approximately 10 million Americans are currently living with mild cognitive impairment (MCI). In fact, 50% of MCI cases are due to AD based on biomarker evidence [8]. Eventually, 15% of MCI patients develop dementia after two years, and one-third develop dementia due to AD within five years [6]. The number of Americans with AD-MCI and AD dementia is approximately 11.2 million [6]. The healthcare cost for AD patients in 2022 is expected to reach USD 321 billion [6]. Furthermore, as the global population ages, the number of people suffering from dementia is expected to triple from 50 million to 152 million by 2050 [9]. Therefore, there is an urgent need to impede or slow down the onset and progress of the disease and ameliorate the disease-debilitating symptoms in hopes of putting an end to this disease and preventing a growing public health crisis [10].

Multifactorial mechanisms are involved in AD pathogenesis, including genetic, epigenetic, biological, and environmental factors networking with each other [11]. However, genomic experiments affirmed that no particular gene could be assigned as a potential target for AD pathogenesis. In fact, multiple genetic and non-genetic factors contribute to disease development [11,12,13,14,15,16,17,18]. In the past decade, there has been an increased understanding of potential targets for disease-modifying therapies that delayed or slowed down the clinical course of AD. According to the U.S. National Library of Medicine’s (NLM) ClinicalTrials.gov Beta (beta.clinicaltrials.gov) [19], there have been 1943 clinical trials of different phases and different designs to investigate new potential molecules in treating AD. Among these studies, 151 clinical trials progressed to Phase III clinical trials, in which safety and monitoring of side effects were investigated.

After decades of AD drug discovery research and billions of dollars spent on clinical trials, we still do not have a single effective anti-AD drug. Promising novel strategies to developing anti-AD drugs keep failing in clinical trials [20,21]. The exact cause of the disease remains the subject of ongoing debate and investigation. At this time, the most plausible hypothesis is that AD is a multifactorial disorder in which genetic and environmental risk factors interact, leading to an acceleration in the rate of normal aging. More than 600 genes contribute to AD pathogenesis, along with environmental factors and epigenetic changes [4,5]. The genetic deficiencies in AD consist of germline mutations, mitochondrial DNA mutations, and single-DNA nucleotide polymorphisms [22,23].

Therapeutic clinical treatment targets of AD aim to enhance behavioral, cognitive, and non-cognitive symptoms of the diseases. In the last two decades, there were no new medication approvals for the treatment or the prevention of AD. Currently, the aim of developing new anti-AD agents is to utilize new disease-modifying agents that delay the onset or slow down the progression of an established disease. Aβ, tau protein, and cell oxidation are the most currently promising targets to modify the pathologic status of Alzheimer’s disease [24]. 

In this review, we aim to provide a comprehensive overview of AD pathophysiological hypotheses, classifications, diagnostic biomarkers, and currently approved pharmacological and non-pharmacological treatment options. Moreover, we discuss the emerging role of the microbiome as well as epigenetics in the pathogenesis of AD and discuss the potential of epigenetic and genetic treatment options. We also provide a summary on drugs being developed or currently investigated in clinical trials, their success, failures or drawbacks, newer target approaches, and most important considerations for conducting effective future clinical trials. In addition, we discuss novel approaches for the proper diagnosis and drug-development for AD through exploiting network biology, artificial intelligence (AI), and machine learning (ML). 

## 2. Alzheimer’s Disease Pathophysiology 

There are two main pathological hallmarks of AD: (1) the presence of increased extracellular Aβ plaques, formed due to the aggregation and impaired clearance of Aβ oligomers (hydrophobic Aβ aggregates) [25,26], and (2) the formation of NFTs [27] that are composed of an insoluble intracellular hyperphosphorylated microtubule-associated protein tau [28,29,30,31,32,33,34,35,36,37,38]. MetaCore’s [39] pathway map presented in Figure 1 summarizes the most important processes, genes, and proteins linked to AD pathophysiology. We further highlighted which network objects on the map that are known AD biomarkers according the MetaCore^TM^, and which ones are drug targets for known drugs (i.e., as a proof of their druggability).

### 2.1. The Amyloid Hypothesis

It was long thought that polymorphism in the amyloid precursor protein (APP) gene (*APP*) causes alterations within and around the Aβ domain of translated APP proteins, altering APP proteolytic processing, and leading to changes in Aβ cleavage and aggregation [41]. This would favor the amyloidogenic pathway, causing the accumulation of Aβ, which contributes to AD pathogenesis [42].

#### 2.1.1. APP

The APP protein contains the Aβ domain, which, in healthy individuals, is cleaved and processed into Aβ and other Aβ-related peptides [43,44,45,46,47,48,49] by α-secretase followed by γ-secretase and later cleared [26,50,51]. The APP protein is encoded by the APP gene (Figure 2), which has 17 transcripts (splice variants), 279 orthologues, 2 paralogues, and is associated with 9 phenotypes (ABeta amyloidosis Arctic type, ABeta amyloidosis Dutchr type, ABeta amyloidosis Iowa type, ABeta amyloidosis Italian type, ABetaA21G amyloidosis, ABetaL34V amyloidosis, ADs, cerebral amyloid angiopathy APP-related, and early-onset autosomal dominant AD). 

APP genetic variants are often observed in Familial Alzheimer’s disease (FAD) [52,53,54,55,56,57,58]. Such mutations in the amino terminus of the Aβ domain result in the production of better substrates for β-secretase (BACE1) activity [55], leading to significant increase in Aβ production and Aβ fragments consisting of 40 or 42 amino acid residues (Aβ40 and Aβ42, respectively) [51,59,60]. These Aβ fragments are formed via the amyloidogenic pathway in AD patients, as BACE1 cleaves APP followed by γ-secretase [50]. Enhanced aggregation of altered Aβ is observed due to mutations affecting the mid region of Aβ [57,58]. Amino acid mutations located beyond the carboxyl terminus region of Aβ increase the production of Aβ42 [55,56], which is considered the major neurotoxic Aβ species [25,42,50,61]. Aβ42 is of particular interest; it is more hydrophobic, more pathogenic, and it aggregates faster than Aβ-40, resulting in an earlier onset of AD.

Furthermore, APP has a substrate inhibitory domain (ASID) that negatively regulates the activity of γ-secretase by decreasing Aβ production [62]. Alterations of ASID region by deletion or single-nucleotide polymorphism were reported, in which an inhibitory effect is reduced, causing an increase in Aβ generation [50,62]. Presenilin (PSEN), the catalytic subunit of γ-secretase [63,64,65,66], may also be modified via mutations in PSEN1 and PSEN2 genes (*PSEN1* and *PSEN2*, respectively) [67], resulting in the destabilization of the γ-secretase–APP interactions [68], causing the generation of longer and more hydrophobic Aβ peptides that contribute to the pathogenesis of AD [41,68,69,70]. 

#### 2.1.2. Tau

Tau is a microtubule-associated protein, found in axons of neurons [71], and it functions as a microtubule stabilizer that enables cytoplasmic extensions for the eventual formation of the axon and the dendrites [72,73]. Once tau is phosphorylated, its ability to bind to microtubules is reduced [74,75,76,77] and self-assembly into tangles/filaments is induced [78], hence affecting its normal function. It is considered as a main pathological lesion in the brains of patients with AD in addition to Aβ fibrils. In fact, AD is considered one of the neurodegenerative diseases that are termed “tauopathies” [79,80,81], where tauopathy indicates a tau-related pathology (i.e., dysfunction and/or tangle formation that contributes to the pathogenesis of AD and other neurodegenerative diseases) [27]. As for AD, this is driven by the intracellular aggregation of tau proteins via different mechanisms. Tau’s phosphorylation can lead to aggregation via liquid–liquid phase separation mechanism [82]. Cleavage of tau by several proteases can also lead to aggregation [83,84,85]. It is also suggested that tau pathology could be a downstream effect of Aβ aggregation or both could enhance each other’s toxic effects [41,86,87]. In vivo, Aβ plaques enhanced the aggregation of human-derived pathological tau injected into mouse brains [88]. In autopsy specimens from deceased AD patients, the number of NFT-positive cells is positively correlated to AD stages [89,90,91,92,93]. 

#### 2.1.3. APOE, TERM2, SORL1, and ABCA7 Mutations

Mutations of genes, such as apolipoprotein E (*APOE*) and triggering receptor expressed on myeloid cells 2 (*TREM2*) [70], would affect the microglial clearance of Aβ. Briefly, APOE protein, which is a lipoprotein, binds to Aβ plaques forming Aβ-lipoprotein complexes that are engulfed into microglia via TREM2 receptors [94]. Mutations of *APOE* and *TREM2,* resulting in mutated proteins, such as APOE2, APOE4, or TREM2 mutations (R47H, R62H, and D87N), would contribute to AD development [94,95,96,97]. Additionally, genetic mutations of *SORL1* and *ABCA7* are also considered as causative or strong risk-increasing variants for AD [70]. Sortilin-related receptor L1 (SORL1) controls APP cleavage and APOE uptake, while increasing ATP-binding cassette transporter A7 (ABCA7) increases Aβ phagocytosis [67]. Hence, mutations resulting in alterations of SORL1 and ABCA7 would act as a risk factor for AD [67]. Other genetic mutations that have low, medium, or high risk for AD are summarized by Scheltens et al. [41,70]. 

### 2.2. The Cholinergic Hypothesis

In addition to the amyloid hypothesis described above, other hypotheses have been suggested such as the impairment of cholinergic function, also known as the cholinergic hypothesis [98]. The cholinergic deficit in AD patients was confirmed in various studies, which highlighted a major deficit in the enzyme responsible for acetylcholine (ACh) synthesis (choline acetyltransferase) [99,100,101], reduced choline uptake [102], reduced ACh release [103], and loss of ACh (cholinergic) neurons [104]. Along with the role of ACh in learning and memory [105,106], degeneration of cholinergic neurons and loss of cholinergic neurotransmission contributes to the worsening of cognitive function in patients with AD [106,107].

### 2.3. The Mitochondrial Cascade (Oxidative Stress) Hypothesis

Neuronal mitochondria are considered the main organelles responsible for the neuronal oxidative stress via the generation of free radicals through their electron transport chain [108]. In the case of high levels of ATP and diminishing electron transport effect, superoxide would be formed from oxygen via mitochondrial respiration [109]. Superoxide would then be converted to hydrogen peroxide by superoxide dismutase and later to hydroxyl radicals and anions by the Fenton reaction [110]. These reactive oxygen species (ROS) would affect redox imbalance, cause neurotoxicity and genomic instability, transcription of pro-inflammatory genes, and cytokine release [111]. ROS would further damage and inactivate parts of the mitochondrial electron transport chain, leading to the formation of superoxide from the electron reduction of oxygen in a positive feedback cycle [109,111,112].

ROS damage mitochondrial DNA (mtDNA) that leads to neuronal functional impairments and damaged mitochondria due to this oxidative stress will not be degraded by mitophagy [108,111,113]. Usually, oxidative stress acts as a signal for mitophagy process to degrade damaged mitochondria, as oxidative stress reduces mitochondrial membrane potential [108]. However, ROS alter parkin, which is a key mitophagy regulator, and inhibits its function, causing the continual presence of mitochondria [113]. Collectively, ROS, DNA damage, and mitochondria contribute to the aging process [114,115]. Briefly, following DNA damage via ROS, kinases and PARP are activated, leading to the decrease of NAD+ production, which is essential for metabolic pathways and ATP production. Thus, oxygen consumption and ATP production would be required to increase to meet high energy demand. Mitochondrial coupling would occur, which increases membrane potential, increases free radical formation, and decreases mitophagy. Furthermore, as discussed above, free radicals would further cause DNA damage. 

Denham Harman proposed the free radical theory of aging, in which free radicals are involved in the changes associated with the aging process [116]. It was later confirmed that free radicals are involved in the aging process and advanced age diseases, such as AD [115,117,118]. Furthermore, the effect of ROS on mitochondria supported the theory that relates mitochondria to the aging process and neurodegenerative diseases, such as AD [108]. This theory is associated mostly with the central nervous system since it consumes 20% of the body’s oxygen and is susceptible to oxidative stress [119]. Neurons would have high sensitivity to free radicals since they are non-dividing and post-mitotic cells and cannot be replaced in the event of damage, leading to mitochondrial dysfunction with aging [118,120]. 

It is observed that mitochondrial dysfunction is prevalent in the aging process [108]. In addition, healthy aging results in reduced mitochondrial metabolism in terms of α subunit reduction of the mitochondrial F1 ATP synthase, thus interfering with ATP production [108]. Eventually, ATP production decreases, ROS production increase, and this will cause an increase in DNA, protein, and lipid oxidation [118,121,122]. Moreover, apart from mtDNA damage caused by mitochondrial dysfunction, nuclear DNA is also damaged, leading to impairments in vesicular function, synaptic plasticity, and mitochondrial function [121].

Excess ROS and bioactive metals, such as copper, iron, zinc, and magnesium are present in AD brains that promote Aβ aggregations and NFT formation [123,124,125,126]. Moreover, the elevated levels of mtDNA oxidation are considered as one of the early markers of AD pathogenesis [118,122]. Moreover, late onset AD pathogenesis may be linked to age-associated mitochondrial decline [108]. The expression and processing of APP would be altered with age-associated mitochondrial decline, leading to the production of Aβ oligomers that aggregate into plaques in AD [127,128]. These Aβ oligomers are associated with neuronal toxicities and ROS generation. Studies have shown that the hydrophobic 25–35 region of Aβ leads to neuronal toxicity and generates ROS, showing that Aβ itself is a source of oxidative stress [108,129,130]. Aβ42 is hydrophobic in nature, and it can reside within the neuronal membrane lipid bilayer and cause lipid peroxidation, identified through 4-hydroxy-2-trans-nonenal (HNE) production that is bound to neuronal proteins [131]. Studies have also shown that the production of HNE—due to lipid peroxidation associated with the residing of hydrophobic Aβ protein in lipid bilayer—and HNE’s neuronal protein binding are linked to cell death [132,133,134]. This are related to neurodegenerative diseases’ pathogenesis, including AD. Furthermore, the oxidative stress triggered by Aβ is also likely to be as result of complexation with redox active metals, such as copper, zinc, and iron [108], which are highly present in AD brains [123,124,125,126], promoting Aβ aggregation into plaques [108]. Copper forms the most stable complex and can generate superoxide and hydrogen peroxide that contribute to AD pathogenesis [127,135]. 

### 2.4. Other Factors Affecting Disease Pathogenesis

Other factors that affect the clinical development of AD include vascular pathology of blood–brain barrier that result in leakage and cause dementia [136,137]. Moreover, elevated iron levels in brain and dysregulation of its metabolism are also present in AD patients as it is linked with cellular damage and oxidative stress [138]. Exosomes may play a part in the spreading of Aβ plaques and NFTs through the brain [139].

## 3. Alzheimer’s Disease Classifications

AD is usually classified on the basis of two criteria: age and heredity (Table 1). Based on age, there are two main forms of AD: late onset AD (LOAD) and early onset AD (EOAD). LOAD is diagnosed at age 65 or older [140], although the degenerative process may begin damaging the brain 20 years before symptoms appear [141]. This is the most prevalent form of the disease, comprising 95% of patients [142]. Polymorphism in *APOE* is a major risk determinant of LOAD, contributing to 60–80% of the disease pathogenesis [140,143,144]. *APOE* encodes for the principal lipid transport protein in the central nervous system (CNS), which has three alleles (E2, E3, and E4). APOE4 surges the risk of LOAD development [11]. Genome-wide association studies (GWAS) identified >20 LOAD risk genes linked to endocytosis, inborn immunity, and lipid metabolism [143,145]. EOAD is typically diagnosed between 40 and 50 years of age, comprises 5% of the total AD cases [146], is linked to a defect in chromosome 14 and myoclonus [147,148], and, therefore, it is referred to as a distinctive autosomal dominant inherited form of AD [140]. Usually, people with Down’s syndrome are more prone to EOAD [149]. In fact, mutations in *APP* genes, particularly *PSEN1* and *PSEN2*, are associated with EOAD development [140]. These *APP* variants explain 5–10% of the autosomal dominant inherited cases, leaving the majority of cases totally unexplained [150,151].

Based on heredity, AD can be either familial or sporadic [152]. Compiling the two diagnostic criteria (i.e., age and hereditability), AD can be classified into EOAD, LOAD, early onset familial (eFAD), late-onset familial, early onset sporadic, and late-onset sporadic. Yet, these classifications overlap in terms of nomenclature, diagnosis, and treatment. FAD is linked to genes [153].

**Table 1 diagnostics-12-02975-t001:** Main types of Alzheimer’s disease based on age and heredity.

Classification	Genetic Factors	Age Onset	Clinical Features	Risk Factors	Top Treatments	References
Early-onset	Yes	40s–50s	Plaques of amyloid and tau proteins	Family history	Acetylcholinesterase inhibitors (Donepezil, Galantamine, and Rivastigmine)	[154]
Late-onset	Yes (APOE)	≥65	(APOE) ε4 allele	Age ≥ 65 years, genetic and environmental factors	Acetylcholinesterase inhibitors (Donepezil, Galantamine, and Rivastigmine) and treatment of vascular risk factors and sleep and mood disorders	[155]
Familial	Yes (PSEN1, PSEN2, APP)	40s–50s	Mutations in PSEN1, PSEN2, and APP	Family history	Acetylcholinesterase inhibitors (Donepezil, Galantamine, and Rivastigmine)	[146,156,157]

## 4. Alzheimer’s Disease Diagnosis

AD is a dual clinicopathological condition, which means that two requirements must be met for the definite diagnosis: (1) the presence of a clinical phenotype characterized by symptoms, such as episodic memory impairment or involvement of other cognitive, behavioral, and neuropsychiatric domains and (2) the development of neurological changes, such as the accumulation of NFTs and Aβ plaques in the brain. NFTs and Aβ plaques can be detected only through autopsy; as such, clinical diagnosis relies heavily on the observation of behaviors that are compatible with known clinical features of AD [41,158] and the exclusion of other potential causes [9]. The guidelines of neuropathological assessment of autopsy samples for the definitive diagnosis of AD have been published by the National Institute on Aging and the Alzheimer’s Association in 2012 [159]. 

In fact, it remains challenging to discriminate AD from other neuropathological dementia despite the advances in research protocols and current diagnostic tools [11]. Currently, AD diagnosis is based on confirming memory loss and cognitive impairments using neurological tests, such as the Montreal Cognitive Assessment (MOCA) [160] and Mini-Mental Status Examination (MMSE) [161]. However, the ultimate AD diagnostic protocol can only be performed post-mortem to detect Aβ and tau NFTs in brains of deceased patients [11]. The current limitations in AD diagnostics burden the development of effective AD treatments since it depends on signs and symptoms, whereas the accurate status of the brain can only be assessed post-death [162]. Currently, scientists are suggesting epigenetics alterations could be exploited as diagnostic surrogates for AD [7,11,12,13,14,15,16,18,163,164,165,166,167,168,169,170,171]. 

## 5. Epigenetic Changes in Alzheimer’s Disease

Epigenetic modifications have emerged as significant contributors in AD pathogenesis, mediating promises for AD treatment [7,11,12,13,14,15,16,18,163,164,165,166,167,168,169,170,171]. Various epigenetic changes, including mitochondrial epigenetics (i.e., mitoepigenetics), DNA methylation and hydroxymethylation, noncoding RNA translation, and histone post-translational modifications have been implicated in AD development [11]. Disruption of both DNA methylation and DNA hydroxymethylation processes has been implicated in many diseases that are classified as neuropathologies, including AD [172]. Interestingly, the key genes involved in AD pathogenesis are regulated by miRNAs and DNA methylation [11,13]. Other studies reported that there is an overlap between distinctively methylated DNA spots in AD and histone signatures in H3K27me3 and H3K4me3 in the Polycomb-repressed (poised) promoter [173]. Further studies declared that indistinguishable 5-methylcytosine (5mC) models are observed in AD patients associated or not associated with schizophrenia [174]. 

### 5.1. DNA Methylation

DNA methylation retains fundamental cellular functions and synaptic elasticity in the CNS, and it influences cognitive processes [175]. DNA hydroxymethylation is essential for neurodevelopment and is concentrated in the CNS, which further signifies the importance of DNA methylation [175]. Some studies showed that overall DNA methylation is decreased in AD patients [176,177,178,179], while other studies recorded no significant differences in DNA methylation between AD and age-matched healthy individuals [180,181]. DNA methylation patterns that were in connection with AD were investigated for the following genes: glycogen synthase kinase 3 beta (*GSK3b*) [182,183], ankyrin 1 (*ANK1*) [184], *TREM2* [17], and brain-derived neurotrophic factor (*BDNF*) [185]. *ANK1* methylation has been increased in AD patients [186,187,188,189]. An increase in DNA methylation has been also recorded in the dorsolateral prefrontal cortex [187], entorhinal cortex [190], temporal cortex [190], temporal gyrus [191], and the hippocampus. Contrarily, a decrease in DNA methylation has been reported in locus coeruleus, prefrontal cortex [192,193], and blood samples [194]. Additionally, studies showed that 13% of noncoding RNA CpG motifs were methylated in AD patients, leading to a significant increase in 5mC levels in these genetic loci in particular [190]. 

### 5.2. Mitochondrial DNA Methylation

Mitochondria generate the energy (ATP) required to vitalize the cell’s reactions, and AD has been postulated to be associated with energetic decrease arising from mitochondrial disorder. Dysfunction of the mitochondrial oxidative phosphorylation and energy-producing cascade increases in reactive oxygen species (ROS) generation and apoptosis such that both are implicated in neurodegeneration and disease development [11,12,13,16,195,196,197]. Studies reported that multiple considerable deletions are detected in mitochondrial DNA (mtDNA) and are linked with AD pathogenesis [198]. In addition, mutations of mitochondrial rRNA and tRNAs [199,200], cytochrome C oxidase [200,201], and the regulatory D-loop influence mtDNA copy number, transcription, and translation [202]. Low levels of mtDNA were observed in AD patients having low Aβ and high tau in the cerebrospinal fluid (CSF) and in presymptomatic patients having *PSEN1* mutation [203]. Low mtDNA copy number and abnormal propagation of mitochondria were associated with low level of mtDNA in CSF that might act as a biomarker for AD in the preclinical phase [203]. Another study demonstrated a positive correlation between Aβ and CSF mtDNA content but a negative correlation between phosphorylated tau protein and CSF mtDNA levels [204]. Low levels of CSF mtDNA accompanied by low Aβ and high phosphorylated tau assist in distinctive AD diagnosis against other neurological disabilities [204]. Studies revealed an increase in methylation of mtDNA at CpG and non-CpG repeats of D-loop of AD entorhinal cortex with Braak stages I to II and III to IV [192,205]. However, a significant decrease in mtDNA methylation in AD blood samples was detected [197]. 

### 5.3. DNA Hydroxymethylation

Studies revealed that thousands of distinguishable hydroxymethylated regions (DhMRs) in AD brains are associated with an increase of 5hmC levels in intragenic regions [206,207,208]. Genomic studies reported an increase in 5-hydroxymethylcytosine (5hmC) levels in the F-box and leucine rich motif protein 16 (FBXL16) gene [186]. *FBXL16* was reported as a potential AD-associated gene, showing an encoding decrease in microglia cells of mouse AD models [209]. Another study reported a decrease in 5hmC levels in four CPG repeats in ANK1 [186]. Diverse studies confirmed a decrease in 5hmC levels in the AD entorhinal cortex, cerebellum [173], and CA3 region of the hippocampus [210] but an increase in 5hmC levels in AD brains’ parahippocampal gyrus [211], middle frontal gyrus, and middle temporal gyrus [212]. Further studies reported an increase in tau protein deposition and a decrease in astrocytes location [213], whereas one investigation recorded that 5hmC is not localized in AD cerebellum and entorhinal cortex [214]. Another study declared a decrease in 5hmC deposition in AD glial cells of hippocampus CA1 region [210]. Further study investigated the effect of *TREM2* on AD pathogenesis and found there is a positive association between 5hmC repeats in exon 2 of *TREM2* and *TREM2* expression, postulating that an increase in gene expression might assist in tissue repair [215]. *TREM2* is encoded in microglia cells and is required in tissue repair, homeostasis, and natural immunity reaction [215].

### 5.4. Histone Modifications

Histones (H1, H2A, H2B, H3, and H4) are biochemically highly basic proteins rich in arginine and lysine residues. Histones serves as a scaffold, assisting DNA to wrap and condense in eukaryotic nucleus forming nucleosomes [216,217]. Histone modifications are involved in neuronal differentiation and growth, older individuals’ brains homeostasis, and in AD pathology [7,11,12,13,14,16,18,164,165,169,171,204,218]. A prevalent lack of heterochromatin was detected in human AD, tau transgenic Drosophila, and mice [219]. Oxidative stress and DNA deterioration were associated with transgenic tau expression and heterochromatin relaxation [219]. 

Histone modifications, such as abnormal acetylation, were linked with aberrant signaling, apoptosis, inflammation, immunity, and neuroplasticity [220]. Histone acetylation was detected in postmortem AD brains [221,222,223]. A decrease in histone acetylation was observed in AD temporal lobes [224]. Acetylation of lysine 16 on histone H4 (H4K16ac) is implicated in aging and DNA damage, and such deterioration was previously observed in AD cortex patients [218,225]. However, acetylation of lysine 12 on histone H4 (H4K12ac) was accompanied with memory disturbance [218]. Higher H4K12ac content was detected in MCI but not detected in AD, confirming its role in an infant stage of disease development and aggregation deposition [15]. High levels of acetylated histone as well as H3 and H4 were detected in human post-mortem AD brains [225]. In addition, higher levels of histone deacetylases (HDACs), particularly class I (HDAC2 and HDAC3), were observed in AD brains’ regions that are involved in memory, learning, and neural plasticity. HDACs are linked with cognitive impairments and synaptic functions [171,218]. On the contrary, other investigations declared a decrease in HDACs in dysfunction brains’ regions that are associated with MCI symptoms [11]. In addition, class II HDACs are implicated in AD pathogenesis [226]. An increase in HDAC6 level was discovered in AD brains’ cortex and hippocampus and in AD animal models [226]. HDAC6 influences tau phosphorylation and degradation as well as tubulin acetylation, and it mediates inflammatory processes [169,227]. A decrease in HDAC6 level results in higher clearance and decrease of tau aggregation and consequently assists in nerve survival [220,228], while an increase in HDAC6 level leads to a decrease in α-tubulin acetylation and subsequently disrupts microtubules’ homeostasis and mitochondrial as well as vesicular transport [220,228]. A decrease in HDAC4 content, another member of class II HDACs, adversely influences learning and memory development [229]. HDAC4 is involved in neural function, and its increase results in apoptosis, whereas its decrease inhibits nerve cell death [229]. Class III HDACs, sirtuins (SIRTs), are involved in synaptic elasticity and memory functions as well as AD pathogenesis [227]. Studies reported that levels of SIRT1 are reduced in the parietal cortex, whereas SIRT1 levels in AD cerebellum are not reduced [171]. Such expression aberrations, probing Aβ and tau deposition, as well as acetylation of lysine 28 of tau protein, lead to tau aggregation [171,220]. Aberrations in histone methylation have been detected in AD patients as well [230]. The levels of histone methyltransferases (HMT) and histone demethylases are significant for brain vitality and memory function [230]. Studies declared that an increase in trimethylation of lysine residue on histone H3 (H3K9), a biomarker of gene silencing and heterochromatin condensation [231], and overexpression of histone lysine methyltransferase 1 (EHMT1) are observed in post-mortem AD brains [232]. The G9a HMT, an enzyme responsible for demethylation of lysine 9 on H3 (H3K9), is involved in cognitive function in mice; however, H3K4 demethylase contributes to human memory deficiency [230]. Studies observed that an increase in phosphorylation of serine 10 on H3 (H3S10), detected in AD hippocampal neurons [233], and an increase in phosphorylation of serine 139 on H2AX, detected in AD astrocytes, might serve as indicators of DNA damage [234]. ADP ribosylation of H1 was detected in AD brains [235]. Altogether, these results shed light on histone aberration in AD pathogenesis and motivate more researchers to explore the complexity of such factor.

### 5.5. MicroRNA 

Diverse microRNAs (miRNAs) target genes are implicated in AD pathogenesis [11,12,13,14,16,18,169,171]. There are approximately 161 miRNAs that could contribute to AD pathogenesis, while ten miRNAs have been linked to AD, including miRNA-9, miRNA-29, miRNA-34, miRNA-107, miRNA-125, miRNA-132/-212, miRNA-146, miRNA-155, miRNA-181, and miRNA-206 [190]. Additionally, specific miRNAs were related to myelin sheath formation and others were involved in AD development, such as SIRT1, BACE1, and APP [190]. miRNAs are also involved in APP degradation and Aβ metabolism by modulating the activity of APP-degrading enzymes, such as BACE1 [236]. Furthermore, many miRNAs were found to regulate BACE1 expression, such as miRNA-124, miRNA-135b, miRNA-195, miRNA-15b, miRNA-29c, and miRNA-399-5p [237,238,239,240]. Other miRNAs, such as miRNA-219, regulate microtubule-associated protein tau (MAPT) gene (*MAPT*), while others, such as miRNA-124-3p and miRNA-125b, modulate the activity of kinases that are involved in the phosphorylation of tau protein [183,241,242,243]. 

BDNF, or abrineurin, expressed by *BDNF*, is a potential regulator of synaptic elasticity and transmission that induces miRNA-132 expression [244]. Studies reported that miRNA-132 and miRNA-212 encoding is suppressed in the early AD stage [245,246]. Other studies declared that miRNA-9 modulates neural progenitor cells’ growth, differentiation, and migration [247,248]. In addition, miRNA-9 upregulates ACE1 [249], and subsequently, increases Aβ formation and accumulation [250].

It was found that downregulation of miRNA-9 modulates calcium/calmodulin-dependent protein kinase kinase 2 (CAMKK2) expression [251], resulting in an increase in phosphorylated tau and Aβ deposition through CAMKK2-cyclic adenosine monophosphate-activated protein kinase (AMPK) cascade [183,252]. Modulation of BACE1 encoding is also carried by miRNA-29, implying that an increased level of BACE1 is associated with a decrease in the level of miRNA-29 [253,254]. Studies declared that miRNA-29 regulates neuron navigator 3 (*NAV3*) that is overexpressed in AD frontal cortexes [255]. Studies showed that both miRNA-34 and tau mRNA are upregulated in AD, suggesting a linked mechanism for AD pathogenesis [256]. Studies demonstrated that miRNA-107 expression is suppressed in AD CNS and blood, particularly at the begging of AD. Further studies exhibited negative association between miRNA-107 expression and *BACE1,* inferring that *BACE1* mRNA could modulate miRNA-107 [257]. In addition, miRNA-107 regulates cyclin-dependent kinase 5 (*CDK5*) that is responsible for CNS integrity and function [258]. Studies showed that higher levels of miRNA-125 stimulate tau hyperphosphorylation, resulting in promoting mitogen-activated protein kinase/extracellular signal-regulated kinases (MAPK/ERK) signaling and increasing p53 expression [183,241]. Studies revealed that miRNA-132/-212 was linked with cognitive function and was suppressed in AD brains [259]. Studies reported that miRNA-146 expression is modulated by nuclear factor kappa-B (NF-κB), and the overexpression of miRNA-146 paves the way for NF-κB to downregulate the translation of complement factor H (CFH) and subsequently influence the inflammatory reaction in CNS [260]. 

Studies showed that the overexpression of some microRNAs (miRNA-155, miRNA-146, and miRNA-124) is associated with over production of APP and Aβ [261]. Studies revealed that miRNA-181 was suppressed in AD CNS [262]. Further investigations showed that the downregulation of miRNA-181 is associated with higher level of Aβ expression [262]. Furthermore, the downregulation of miRNA-181 influences MAPK signaling cascade [262]. Other investigations reported that miRNA-206 is overexpressed in AD CSF and blood [51,263]. 

## 6. Alzheimer’s Disease Biomarkers

Biomarkers are important tools for the accurate diagnosis of many diseases, including AD. Despite the recent advances in diagnostic methodology for Alzheimer’s disease, differentiation of Alzheimer’s dementia from other forms of dementia remains challenging. The analysis of Aβ-42, total tau protein, and phosphorylated tau (p-tau) from cerebrospinal fluid (CSF) is currently considered the best-established biological marker for the diagnosis of AD as well as differentiation from mild cognitive impairment and other types of dementia. The familiar AD biomarkers are the reduced levels of Aβ in CSF and the appearance of Aβ or tau depositions in the brains of AD patients [264,265,266]. Additionally, biomarker evidence obtained through PET can be used to attribute the clinical syndrome of dementia or MCI to underlying AD pathology, with varying probability [264]. In most cases, AD diagnosis in living patients continues to rely on the patient’s clinical history, family members with neuropsychological conditions, and the observance of symptom progression over time [41].

Before the early 2000s, the only sure way to know whether a person had AD or another form of dementia was after death through autopsy. Today, we have 12,073 biomarkers linked to AD. An overview of these biomarkers is provided in Figure 3a–d. Approximately 441 biomarkers are either approved or in late-stage clinical studies for AD diagnosis, prognosis, staging, and monitoring of disease progression. The most widely used AD biomarkers are Aβ42 (the major component of amyloid plaques in the brain), tau, and phospho-tau (major components of tau tangles in the brain) [267]. These biomarkers are measured in CSF, which is the clear fluid that surrounds the brain and spinal cord, providing protection and insulation.

In May 2022, the US FDA authorized the use of Lumipulse G beta-Amyloid Ratio (1-42/1-40) in vitro diagnostic test for the assessment of beta-amyloid pathology in CSF samples [268]. The ratio of these two proteins in CSF is indicative of the presence of amyloid plaques. The test is minimally invasive, and it is the first FDA-authorized in vitro diagnostic biomarker for use in individuals being evaluated for AD and other causes of cognitive decline. However, results of the test must be interpreted in conjunction with other patient clinical information. Additionally, Aβ42 levels measured in plasma have been evaluated as a potential biomarker for AD since it is less invasive to sample plasm than CSF. All recommended/approved AD biomarkers for disease diagnosis and prognosis are list in Table 2.

## 7. Anti-Alzheimer’s Drugs

There are currently 868 anti-Alzheimer’s drugs in different stages of development. However, only 273 drugs are currently under active development by biotech and/or pharma with evidence of active development in the last 6 months, according to Cortellis Drug Discovery Intelligence [269]. The most effective drugs currently approved for AD management are listed in Table 3, and they comprise cholinesterase inhibitors (donepezil, rivastigmine, and galantamine) and the *N*-methyl-D-aspartate (NMDA) receptor antagonist (glutamate antagonist) memantine [41,70]. All of these drugs offer symptomatic treatments.

### 7.1. Drugs under Active Development 

There are currently 273 drugs under active development for the treatment of AD, including small molecules, biotechnology products, peptides, combinations, and herbal materials (Figure 4a). The majority of these drugs target Aβ42 precursor protein (24.9%), followed by APP (18.7%), MAPT (10.6%), acetylcholinesterase (AChE) (3.3%), cholinergic receptor muscarinic 1 (CHRM1) (3.3%), NMDA receptor (2.2%), tumor necrosis factor (TNF) (2.2%), 5-hydroxytryptamine receptor 6 (5-HTR6), cholinergic receptor muscarinic 4 (CHRM4) (1.8%), glucagon-like peptide 1 receptor (GLP1R), insulin (1.5%), sigma non-opioid intracellular receptor 1 (SIGMAR1) (1.5%), sodium channel (1.5%), and 5-hydroxytryptamine receptor 4 (5-HTR4) (1.1%). Drug count details are shown in Figure 4b. Additionally, the top organizations developing these drugs as well as the development status of these drugs are shown in Figure 4c and Figure 4d, respectively.

Recently, monoclonal antibodies (mAbs) have revived the hope for AD treatments. Aducanumab, an mAb, targets Aβ aggregates in AD patients’ brains to decrease their formation [10,277]. In 2021, aducanumab was approved for AD and prescribed for individuals with AD-MCI and mild AD dementia [10,277]. It is a humanized recombinant monoclonal antibody to Aβ. In a clinical study on 165 patients, aducanumab demonstrated significant reduction of soluble and insoluble Aβ. Furthermore, aducanumab reduced AD clinical decline measured by Mini-Mental State Examination scores. At 12-month follow-up, cerebral Aβ disappeared from almost 50% of patients diagnosed with mild AD. A Phase III clinical trial on 1638 patients of aducanumab has been terminated due to safety and efficacy issues [278]. In addition, two monoclonal antibodies, donanemab and lecanemab, are currently under US Food and Drug Administration (FDA) investigations [277,279,280,281]. 

The pro-drug of methylene blue, leuco-methylthioninium, is a second-generation tau aggregation inhibitor (TAI) and the only tau-specific agent to undergo Phase III clinical trials. Two Phase III clinical trials were conducted in 2016 to demonstrate the efficacy of different doses of leuco-methylthioninium and to compare the efficacy of monotherapy compared with combination with cholinesterase inhibitors or memantine. A third clinical trial to demonstrate the efficacy of low dose leuco-methylthioninium is still active and recruiting to date [282]. Anti-tau monoclonal antibody (tau vaccine) is an IgG4 antibody that targets aggregated tau protein. Preclinical and Phase I clinical trial data demonstrated that it was safe and might present a potential agent for treating AD. A 96-week Phase II safety and efficacy trial (453 participants with AD) was conducted. Recruitment has completed (in August 2022), but the final study report is not yet published [283]. Gosuranemab is a therapeutic mAb for the N-terminal of extracellular tau. Gosuranemab was demonstrated to be safe and effective in a single ascending dose study. Gosuranemab has been investigated through a Phase II clinical trial (654 participants with MCI or mild AD). However, the study was terminated due to lack of efficacy following the placebo-controlled period readout [284]. Semorinemab is another antibody that targets the extracellular tau. Promising results were concluded from a pilot safety study, and currently, a Phase II clinical trial (272 patients with moderate AD) to investigate cognitive function and functional capacities of patients is still active [285]. Zagotenemab, an mAb, binds to tau aggregates. Phase I clinical trial (single dose) was conducted on zagotenemab in patients with mild AD [277]. 

### 7.2. Withdrawn, Discontinued, or Suspended Drugs

There are 108 drugs that have been either withdrawn, discontinued, or suspended from use for AD. The majority of these drugs were small molecules, but there were some biotechnology products and few peptides (Figure 5a). Many of the drug targets (Figure 5b) are similar to drug targets under active development for AD, which may give the impression that those drug targets may not be successful for the treatment of the disease, especially since many of the big pharmaceutical companies have abandoned them, including Pfizer, Sanofi, Lilly, AstraZeneca, and others (Figure 5c). Table 4 summarizes the major failures and suggests hypotheses explaining them.

## 8. Exploiting Network Biology Approaches in Alzheimer’s Disease Research

Network biology approaches have been suggested as paradigm-changing approaches for the discovery of disease biomarkers, drug targets, and effective drugs for polygenic multifactorial diseases, including cancer, diabetes, psychological disorders, and AD. However, the typical focus on one single type of omics has been a limiting factor for the success of previous systems biology studies because the findings were explaining only a modest portion of the complex disease, and AD was no exception. Therefore, future studies should study multiple omics data simultaneously and apply new technologies, including machine leaning (ML) and artificial intelligence (AI) to derive novel multi-system and multi-target hypotheses. 

### 8.1. Previous Alzheimer’s Disease Drug Discovery Failures

Misunderstanding of the disease mechanisms coupled with inconsistent drug development protocols that relied on single-target approaches, in addition to the improper management of drug discovery projects, led to the inopportune nomination of drug targets which contributed to many drug failures [3,286]. Additionally, clinical trial design utilized in drug discovery failed due to many reasons, including the delay in initiation of treatments, incorrect drug doses, or lack of good drug-monitoring biomarkers [287,288]. The success rate in progressing AD clinical trials from one phase to the next has been poor, and the number of therapeutic agents approaching FDA approval is low [289]. Failures in clinical trials might be due to ineffective treatments, drug side effects, or misconducted trials [289]. The improper selection of methodological parameters in clinical trial design [290] impeded the success of previous clinical trials [287,288,289,291]. In fact, the clinical trials dilemma in psychiatry, neurology, and AD has been discussed elsewhere by many researchers [292,293,294]. Issues including inaccuracy, incorrectness, and bias hindered clinical trials success [295,296]. Other factors included personal errors, drawbacks in rating scales, and limitations in neuropsychological tests leading to errors regarding the underestimation of the clinical outcome in clinical trials [290,297]. Increasing the number of clinical trials investigating drug effects has been associated with better treatment outcome [289].

Additionally, limitations in cell-based models to probe neurodegenerative diseases, such as AD, contributes to AD failure treatment [298]. The complexity of CNS motivates researchers to integrate the molecular basis of neurodegenerative diseases with the unique organization and construction of brain tissue [298]. This combined approach is displayed via 3D cell models accompanied by microfluidic technology, which are in their early stages and ready for improvement [298]. Subsequently, this integrative system should enrich the preclinical drug development pipeline [298]. The biodiversity of AD-drug design and development needs to unify healthcare workers’ and scientists’ efforts [289]. Other factors that played an important part in the failure of many AD drug development programs were improper diagnostic evaluations, elusive genetic factors, and/or concomitant diseases [299,300].

### 8.2. Network Biology Approaches Hold the Promise to Revolutionize Alzhiemer’s Disease Research

AD is a complex disease associated with multiple perturbations in biological networks and functional network connectivity that are fundamental for normal physiological function; hence, multi-target treatment approaches seem imperative to treat the disease [301,302,303]. Studies have reported that numerous brain functional networks are significantly impaired in AD patients, including the control network (CON), default mode network (DMN), dorsal attention network (DAN), salience network (SAL), and sensory–motor network (SMN) [304]. In mild AD patients, there is evidence indicating reduced functional network connectivity in the brain is a predisposing factor. Additionally, the DMN is impaired in very mild to mild AD patients, while severe AD patients suffer from disrupted network crosstalk [304]. Thus, network and systems biology approaches that target multiple disease networks and pathways hold great promise to revolutionize AD drug discovery research. 

Furthermore, network biology approaches enable the identification of novel disease biomarkers, including quantitative diagnostic and prognostic biomarkers, imaging, and biochemical tests. Novel validated disease biomarkers could potentially equip AD researchers with the proper tools to accurately differentiate between AD and non-AD dementias, which can positively impact drug discovery efforts, clinical trial design, and patient selection for clinical trials [305]. There is agreement among scientists [1,2,3,4,5,22,306] that future AD research should focus on the following: (1) reassessing previous and current prevalent AD pathogenesis hypotheses, (2) identifying effective disease-specific biomarkers, (3) re-evaluating previous disease diagnostic standards, (4) considering new guidelines and procedures for disease control, (5) reorienting drug discovery efforts toward employing approved multi-target approaches and pharmacogenetic hypotheses, (6) updating the managerial requirements for drug design and development, (7) applying pharmacogenomics approaches in biomarker and drug discovery and development, and (8) implementing disease-prevention strategies for susceptible individuals. 

### 8.3. Current Network Biology Efforts

The underlying hypothesis of network medicine has been recruited in the development of multi-target ligands and combination drugs [21]. The multi-target ligands and combination drugs are considered promising network medicines for challenging and complex diseases [307,308]. Clinical studies showed that multi-target ligands and combined drugs are more effective than single-target drugs in complex diseases treatment, including depression, cancer, and infectious diseases, such as the acquired immunodeficiency syndrome (AIDS) [309,310,311,312]. Combining donepezil and memantine improve the brain’s cognition function, and patient’s overall status in mild and advanced AD. Additionally, such drugs decrease the rate of clinical decay and are safe and tolerable [307]. 

The idea of network medicine is based on the hypothesis that diseases occur due to the disruption of biological networks responsible for homeostasis as a result of activation or deactivation of certain proteins or biochemical reactions, which eventually disturb the balance of normal physiology pathways [313,314,315]. Hence, disease networks are complex disease processes that are caused by irregular diverse genes, proteins, and signaling cascades [308]. Therefore, network medicines intend to restore disrupted disease networks to their default normal physiology status by targeting multiple key effectors in disease pathways [308]. 

Recent advances in multi-omics data analysis coupled with advancements in computational chemical biology methods led to better disease understanding. As a result, network medicines have been suggested as potential surrogates for identifying effective treatments for complex diseases, including AD [301,302,303,316]. Additionally, the application of network approaches to AD research projects has shed light on a crosstalk among diverse signaling pathways involved in AD pathogenesis [21]. Further work is required to lay the groundwork for the development of the next-generation anti-AD drugs. Furthermore, the diverse disease networks could not be revived through targeting of a single protein and/or signaling cascade because there are numerous active and spare cellular mechanisms in biological systems [317]. 

### 8.4. Multi-Target-Directed Ligands as Network Biology Treatments

The Multi-Target-Directed Ligands (MTDLs) approach is one of the most promising therapeutic interventions for AD patients as well as other complex multifactorial diseases, including cancer, diabetes, and other psychological disorders [318,319,320]. The design of MTDL hypothesizes that successful disease-modifying treatments of AD should target systems biology pathways rather than selectively targeting individual proteins or drug targets [321,322]. As such, MTDLs can be defined as drugs and/or technologies designed to interact with more than one target involved in the pathogenesis of a defined disease [322,323,324,325,326,327,328], surpassing the “one-molecule, one-target” model [313,319]. It has been theorized that potent MTDL should simultaneously target the typical signs of AD, such as the irregular accumulation of Aβ peptides [329,330,331,332,333], tauopathies [334,335,336,337], and the cholinergic insufficiency in CNS [106,338,339]. In addition, the effective MTDL should consider other AD features, such as the oxidative and nitrosative stresses [117,340,341], inflammatory response of brain and spinal cord, excitotoxicity [342], mitochondrial dysfunction [327,343,344], aberrances in calcium [345,346] and other metals [19,267,347], and irregularities in apolipoproteins [348,349].

It is suggested that the rational design of MTDLs can be achieved by two approaches: (1) drug repurposing, considering drug design methods that take into account the biological fingerprints (or biological spectra) of familiar active drugs against other therapeutic receptors where one or more drugs can modulate several targets [269,350,351,352,353] and (2) fragment-based drug design, which is based on the core structures of active compounds against specific targets to generate a new merged scaffold with dual or multiple activity against two or more targets [269]. 

In the first approach, compounds are screened against multiple proteins/drug targets to retrieve hits with the desired biological profiles [350,351,352]. The main advantage of this approach is that the investigated compounds are often commercially available and clinically proven to be safe, thus reducing development time and costs [269,350,351], and most importantly, the proposed lead might act as a synergistic effector, modulating the disease pathway effectively [354]. However, the optimization protocol of the biological activity of the lead compound to fit the new disease application has been limited [269,350,351]. Therefore, more work is required to improve hit identification and lead optimization. Sometimes the pharmacokinetics properties of the lead hinder the application to new diseases such as AD where drugs have to meet the criteria for CNS drug design [355,356]. The latter, fragment-based, MTDL approach can be designed using three main methods: (1) linking active fragments/compounds using a linker/spacer and keeping known pharmacophoric features [269], (2) fusing or integrating the active compounds to generate a new chemical entity that shares identical features [269], and (3) merging/mixing the selected bioactive compounds to yield a scaffold that has the key functionalities of the pharmacophore [269]. 

Studies indicated that the major impedance of the MTDL success is the need to maintain or boost the biological activity of the prioritized compounds while preserving drug-like properties [40]. Many MTDLs may have limitations due to lower selectivity towards some drug targets [357,358,359,360,361,362], while drug development efforts focusing on increasing the biological activity of MTDL may increase the risk of drug toxicity [357,358,359,360,361,362]. Therefore, MTDLs should be optimized by improving the selectivity towards certain protein targets while reducing drug toxicity [359,360,361,362]. Additionally, the designed chimeric entities using the fragment approach have higher molecular weights than the parent compounds, which may affect drug-like properties, while at other times, the merging protocol might be a promising solution for developing oral bioavailable drugs [363,364,365]. 

Finally, when considering MTDL, it is crucial to pay special attention to the required physicochemical properties, including pharmacokinetics, pharmacodynamics, hydrophilicity, and hydrophobicity [269]. MTDL design against neurodegenerative disorders should take into account the drug’s blood–brain barrier permeability [355,356]. 

### 8.5. Suggested Disease Biomarkers and Disease Modifying Drugs

Known diagnostic and prognostic biomarkers for AD [269] significantly enrich pathways involved in inflammation and immune regulation. AD biomarkers can be divided into two groups: 168 EOAD biomarkers [154,366] and 932 LOAD biomarkers [367,368]. There are 69 biomarkers that overlap between EOAD and LOAD: ACO2, ACTB, ACTG1, ADAM10, ADIPOQ, ADRA1A, AIF1, APP, ANG, ACE, APOE, ABCA7, ATP6V1B2, ATP2A2, AURKC, AXL, BACE1, CACNA1G, CD33, CLP1, CLU, CR1, DICER1, DUSP13, DNMBP, FNDC5, GRK5, GBA1, GRN, H3C1, H3C10, H3C11, H3C12, H3C2, H3C3, H3C4, H3C6, H3C7, H3C8, IL1B, IL6, IL6R, KIF5A, HLA-DRA, MTHFR, MAPT, MBP, NSF, NDRG4, NRGN, NCSTN, NSUN2, PAK1, PLD3, PSEN1, RTN3, SLC10A3, SLC12A5, SLC24A4, SORBS2, SORL1, SPARCL1, TCIRG1, TYROBP, TREM2, TNF, YWHAG, VSNL1, and VWA2.

In order to get a better idea of these 69 overlapping biomarkers, we used the compared experiment workflow in Metacore [39] to compare enrichments results in pathway maps for EOAD and LOAD biomarkers. We found that the top enriched pathway map by common disease biomarkers for EOAD and LOAD is “protein-folding and maturation related to angiotensin system maturation” (Figure 6).

LOAD biomarkers led to more significant enrichments of immune system and allergic response pathways, apoptosis, tissue remodeling and repair, cell differentiation, cell cycle regulation, and neurofibromatosis [366], while EOAD led to more significant enrichments of heart failure pathway maps, stem cells, spermatogenesis, lipid biosynthesis regulation, and blood clotting pathways [366]. 

## 9. Artificial Intelligence and Machine Learning Approaches

Machine learning (ML) and artificial intelligence (AI) have been used successfully to extract insight from ‘big’ biological data [369,370,371]. Domain expertise from biology, genetics, elderly medicine, psychiatry, psychology, neurology, and neuroscience could be combined with new bioinformatics and statistical analytical tools to gain insight from multi-omics data. Such insight is valuable for providing answers for challenging research questions, and it can be achieved through the use of theoretical modeling [372,373]. In AD research, ML and AI can answer critical questions about combination diagnostic biomarkers, AD patient subgroups, and disease pathogenesis, thus supporting the identification of a personalized treatments for AD patients [374,375]. In fact, the use of AI has been suggested to probe the pathogenesis mechanisms of AD by analyzing big multi-omics data in parallel [376,377]. Additionally, AI has the capability to differentiate AD patients from other patients suffering from non-AD cognition impairment. It can also anticipate the progression from MCI to AD dementia and assign a tailored treatment for each individual patient [376]. Furthermore, the application of ML and AI approaches to AD research data, can lead to novel hypotheses regarding efficient interventions for AD patients [376]. AI can also aid in the diagnosis of the early stage of dementia [268]. 

Many research efforts focused on utilizing ML and AI approaches to mine data from clinicaltrials.gov records to evaluate anti-AD therapeutics in different stages of clinical development to study their mechanisms of action and important clinical trial characteristics [10,372,378,379,380,381,382]. AI and ML approaches can lead to important discoveries by learning from the recent advances in clinical trials and anti-Alzheimer’s drug development pipelines [383,384,385,386,387,388]. Complex AI-based models could be exploited to inform researchers and health care providers about diverse disease etiologies, effective diagnostic biomarkers [375], and individualized treatments based on network biology approaches [389,390]. 

## 10. Exploring Epigenetic Treatments

Studies showed that DNA methylation/hydroxymethylation is dysregulated in AD patients prior the onset of clinical symptoms [11]. These were presented in a prospective study on autopsied brains, as level of methylation, in terms of 5mC levels, in presymptomatic patients is similar to those with AD patients [189]. The levels of 5mC, 5hmC, and ten–eleven translocation 1 (TET1) proteins were elevated in preclinical AD patients and AD patients compared with the control group [171,211]. Although further validation is required, DNA methylation/hydroxymethylation may be used as a biomarker for AD diagnosis [11]. 

Histone modifications, particularly acetylation, deacetylation, and methylation dysregulation, play a role AD pathogenesis [11]. HDACs are highly expressed in patients with AD [171,218], affecting learning, memory, and cognition; hence, HDAC inhibitors (HDACi) are considered a potential treatment option [391]. Studies on AD patients showing low histone acetylation were reported [211,224,225], allowing the potential use of histone acetyltransferases (HATs) [211]. Increased levels of histone methylation and histone methyltransferase enzyme mRNA were reported in postmortem brains of AD patients [11,218]. Although the loss of histone methyltransferase function would affect learning capabilities in AD patients [11], the use of partial histone methyltransferase inhibitors [211] would restore the balance between histone methylation and demethylation in patients with AD to maintain brain integrity and memory [230]. Inhibitor of histone acetyltransferases (INHAT) is reported to bind to histones and block their access to HATs [392]. Studies showed that ANP32A, which is a component of INHAT and inhibitor of protein phosphatase-2A, is upregulated in AD patients [393,394]. In an in vivo study, the down regulation of ANP32A would reduce INHAT formation and allow for histone acetylation [395]. Collectively, drugs from those classes would comprise potential therapeutic options for AD treatment. 

HDACi are considered to be non-selective [220], but they are beneficial, as they reduce AD hallmarks [225]. The use of HDACi that selectively inhibits HDAC2 and HDAC3 would improve cognition, in contrast to inhibiting HDAC1 that would result in neurotoxicity [11,225]. HDAC6 selective inhibitors were also shown to have neuroprotective effects [228,229]. Sirtuins, which are a class of HDACs, contribute to AD pathogenesis and selective inhibitors would also be beneficial [227,229]. Although some HATs showed better response than non-selective HDACi, their low membrane permeability and solubility limit their use in AD treatment [11]. 

The miRNAs are responsible for the regulation of gene expression through post-transcriptional gene silencing [396]. In relation to AD, several studies summarized by Nikolac Perkovic et al., 2021 [11] showed that miRNAs would be either downregulated or upregulated, altering proteins and enzymes expression responsible for AD pathology. Hence, the use of miRNA mimics to downregulate the expression of genes or proteins [397] or anti-miRNA therapies to alter the function of a specific miRNA [398] are also considered potential treatment options for AD patients. 

## 11. Genetic Treatments

Targeting genetic alterations in AD patients and consequent gene editing and correction is another potential treatment strategy. These include the use of programmable nucleases, such as zinc finger proteins (ZFP), transcription activator-like effectors (TALE), and RNA-guided clustered regularly interspaced short palindromic repeats (CRISPR)/CRISPR-associated protein 9 (Cas9) [399]. The latter showed more promising results for AD treatment and other neurological diseases than did ZFP and TALE [400,401,402]. The presence of the mutant Cas9 protein, dead Cas9 (dCas9), advanced the CRISPR/Cas9 editing tool, resulting in the emergence of CRISPR interference (CRISPRi) and CRISPR activation (CRISPRa) technologies, in which dCas9 is fused or interacts with transcriptional repressors or activators, respectively [403]. With regard to epigenetics, AD, and dCas9 protein, studies showed promising results with targeting histone demethylase [404], histone acetyltransferase [405], and histone methyltransferases [406,407]. 

## 12. Non-Pharmacological Treatment Options and Preventive Measures

Non-pharmacological treatments encompass several recommendations for various lifestyle modifications, including physical and social activity, tobacco cessation, alcohol consumption, weight management, nutrition, and regular exercise. Other interventions include underlying-disease management (e.g., hypertension, diabetes, dyslipidemia, depression, and hearing loss), as stated in WHO guidelines [408]. More studies should assess the relationship between vaccines and AD; it was found that flu vaccines reduce the risk of AD development [409]. However, the protective mechanisms have not yet been elucidated.

Aberrations in the ecosystem of microbiome have been implicated in diverse gastrointestinal and metabolic dysfunction, such as diabetes, insulin resistance, obesity, and inflammatory bowel disease [410]. In addition, studies showed that changes in gut microbiome is associated with neurological disorders, such as multiple sclerosis (MS), autism, and Parkinson’s disease [411,412,413]. Studies recorded a decrease in microbial diversity in gut microbiome of AD patients [414,415,416,417,418,419,420,421]. Further studies in rats suggested that alterations in gut microbiome might proceed Aβ deposition [422].

## 13. Special Considerations for Clinical Trials

Aspects to be considered when designing a clinical trial include trial rationale, outcomes of interest, statistical analysis design, sample size and recruitment, and interim monitoring [423]. Common clinical trial designs include single-arm trials, placebo-controlled trials, crossover trials, and factorial trials [424]. In AD-related clinical trials, infrastructure and technology, cultures and linguistics, regulatory and reimbursement issues, academia and industry harmonization, availability, and access were considered to be the ultimate challenges that limit the conducting of successful clinical trials [425]. 

According to NLM’s ClinicalTrials.gov Beta (beta.clinicaltrials.gov), 109 clinical trials related to AD were terminated in the last ten years [19]. AD clinical trials were terminated due to the following reasons: unavailability of further funding, halted visits due to COVID-19, feasibility of enrolment, safety issues, slow recruitment of eligible participants (patients), inappropriate study design to achieve the trial’s endpoint, new safety or efficacy data from other studies, unfavourable risk–benefit ratio, and inappropriate dosage settings. Yet, patient recruitment remains the ultimate determinant in AD clinical trials. 

Therefore, there is a need for new and advanced clinical trials designs to accelerate passage through the legal authorities’ requirements to register new promising molecules for treatment and/or prevention of AD. However, new investigation approaches need to be fully validated before they can be implemented in clinical trials [426].

## 14. Conclusions

AD is a multifactorial and polygenetic disease. Novel disease diagnostic biomarkers and disease-modifying treatments are required to halt or slow the onset and disease progression, decrease behavioral aberrations, and ameliorate cognition in AD patients. The recent advances in network biology approaches coupled with the advances in clinical trial design and protocols, in additional to the availability of powerful machine learning and artificial intelligence algorithms, hold promise to identify novel diagnostic biomarkers, better drug targets, and effective disease-modifying drugs. Herein, we provide a comprehensive review on Alzheimer’s disease highlighting the mainstream hypotheses explaining disease pathophysiology as well as current disease treatments and drug discovery projects. We also emphasize the recent scientific evidence implicating epigenetic mechanisms and the microbiome in AD pathogenesis and progression. We suggest that the application of Al and ML approaches in analyzing AD network biology derived from AD data, including genetic, transcriptomic, epigenetic, and metagenomic data would revolutionize our understanding of the disease pathways and will lead to the discovery of novel biomarkers and drug targets. Ultimately, these studies will increase our chances of identifying validated diagnostic biomarkers and effective disease-modifying cures. Hence, breakthrough discoveries in AD research are more likely to occur in the near future. This review provides a summary of the current hypotheses regarding AD pathogenesis in addition to the most recent advances in the search of effective disease biomarkers and drug targets. This review also details AD drugs in various stages of development and highlights technologies that are expected to accelerate AD drug and biomarker discoveries.

## Figures and Tables

**Figure 1 diagnostics-12-02975-f001:**
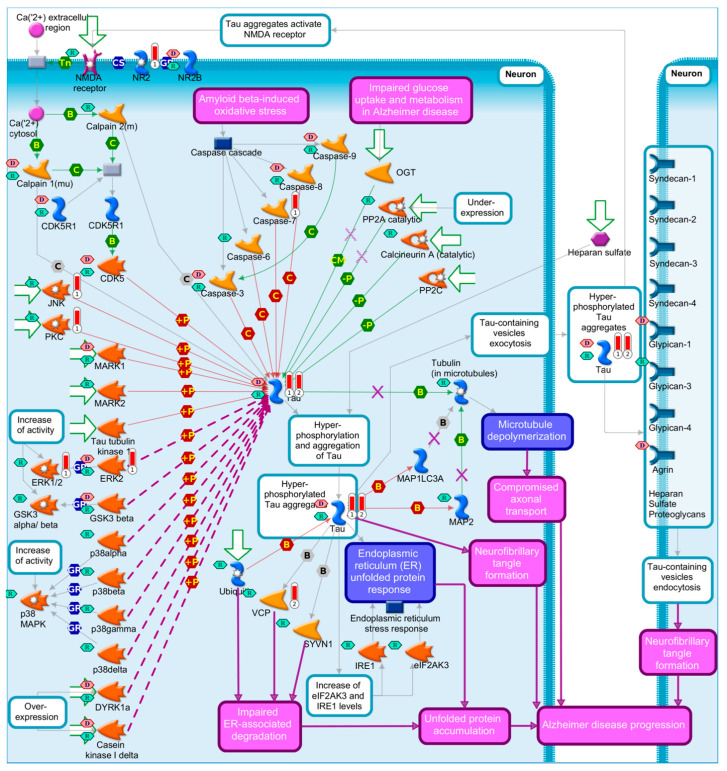
Alzheimer’s disease (AD) pathway map highlighting validated diagnostic/prognostic AD biomarkers in addition to genes/gene products involved in the tau pathology pathway. Connections between network objects on the map are referred to as links (or edges). A link identifies an interaction or a logical relation between two nodes. The type of interaction or relation is reflected by an appropriate symbol placed in the middle of the link. B = binding (i.e., physical interaction between molecules); C = cleavage; CM = covalent modification; +P = phosphorylation; −P = dephosphorylation; Tn = transport; TR = transcription regulation; red arrows = inhibition; green arrows = activation; grey arrows = unspecified action; solid purple arrows = emergence in disease; dashed purple arrows = enhancement in disease; arrows with purple x = disruption in disease; light violet text box = normal process; pink text box = pathological processes; white text box with blue outline = notes; grey block = reaction; blue block = normal process; pink block = pathological process; solid purple hexagon = compound; CS = complex subunit; GR = group relation; starred network objects = groups or complex processes; thermometers on pathway map = network object is also a validated diagnostic or prognostic disease biomarker from the Cortellis Drug Discovery Intelligence (CDDI) database [40]; thermometer 1 is for late-onset AD (LOAD); thermometer 2 is for early-onset AD (EOAD); a pink hexagon with a capital D on the upper left side of the network object indicates an AD biomarker according to MetaCore^TM^; a blue hexagon on the upper left side of the network object with a capital R indicates that the network object (i.e., gene/gene product) is a drug target (not necessarily for AD). Map generated using MetaCore^TM^ version 21.4. MetaCore^TM^, a Cortellis™ solution, 14 October 2022, © 2022 Clarivate. All rights reserved.

**Figure 2 diagnostics-12-02975-f002:**
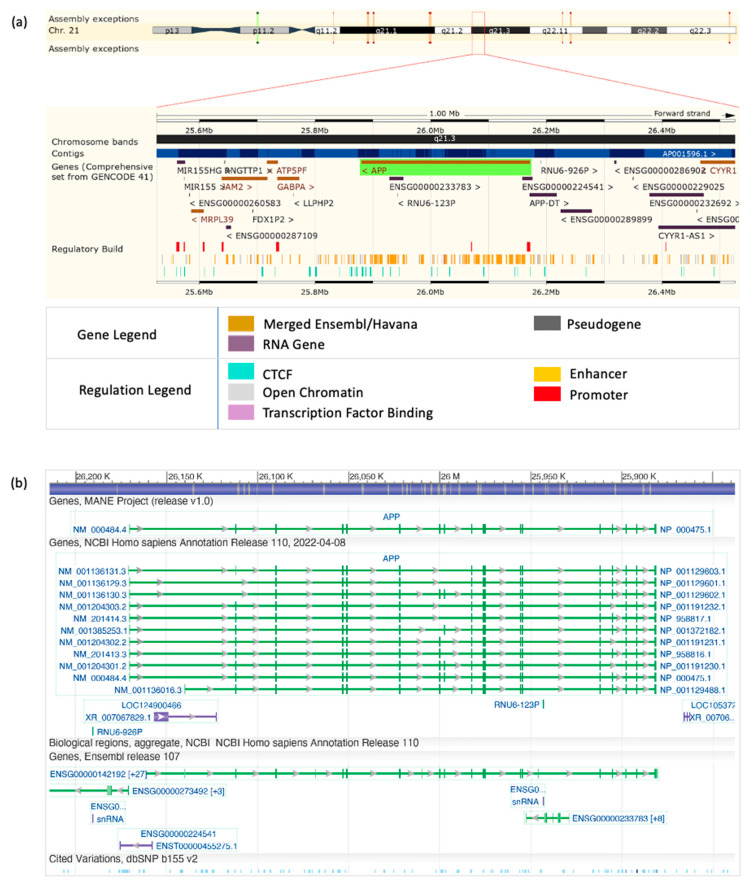
Visualizing APP gene information using (**a**) Ensembl genome browser 107 (July 2022, GRCh38) and (**b**) NCBI’s gene browser.

**Figure 3 diagnostics-12-02975-f003:**
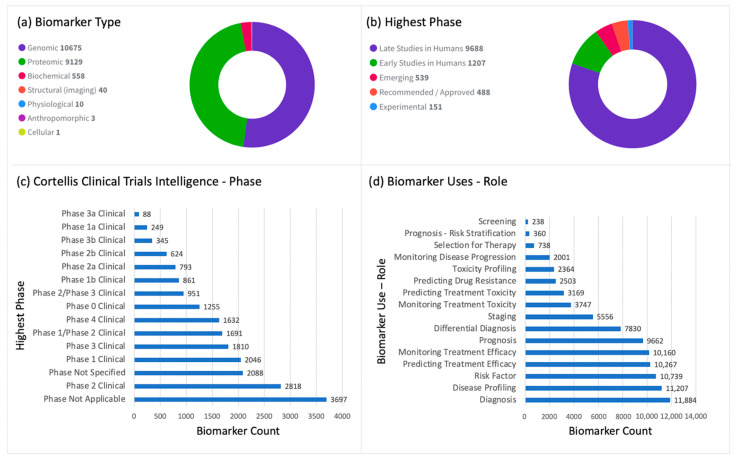
An overview of Alzheimer’s disease biomarkers in all stages of clinical development. Biomarkers are presented by (**a**) type of biomarker, (**b**) highest phase of biomarker development, (**c**) clinical phase of development for clinical biomarkers, and (**d**) according to the biomarker use. Data source: Cortellis Drug Discovery Intelligence, 10 October 2022, https://www.cortellis.com/drugdiscovery/ © 2022 Clarivate. All rights reserved.

**Figure 4 diagnostics-12-02975-f004:**
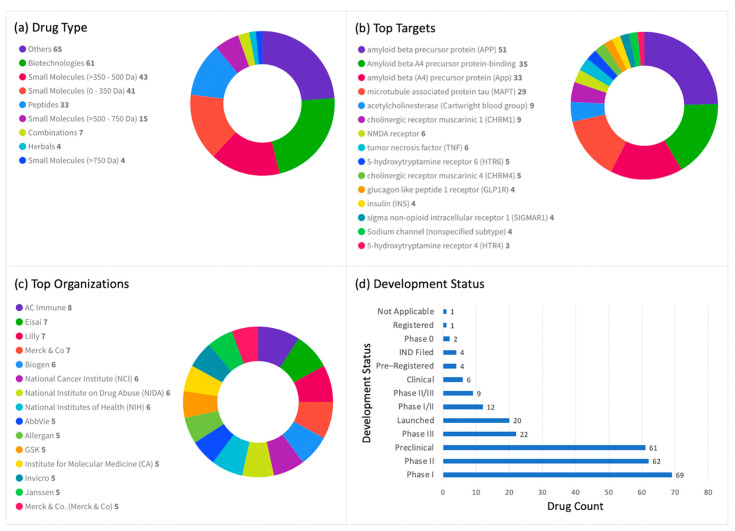
An overview of Alzheimer’s disease drug discovery pipeline under active development by (**a**) drug type, (**b**) top targets, (**c**) top organizations, and (**d**) development status. Under active development, according to Cortellis Drug Discovery Intelligence (CDDI) [40] database as of 10 October 2022. Data source: Cortellis Drug Discovery Intelligence, 10 October 2022, https://www.cortellis.com/drugdiscovery/ © 2022 Clarivate. All rights reserved.

**Figure 5 diagnostics-12-02975-f005:**
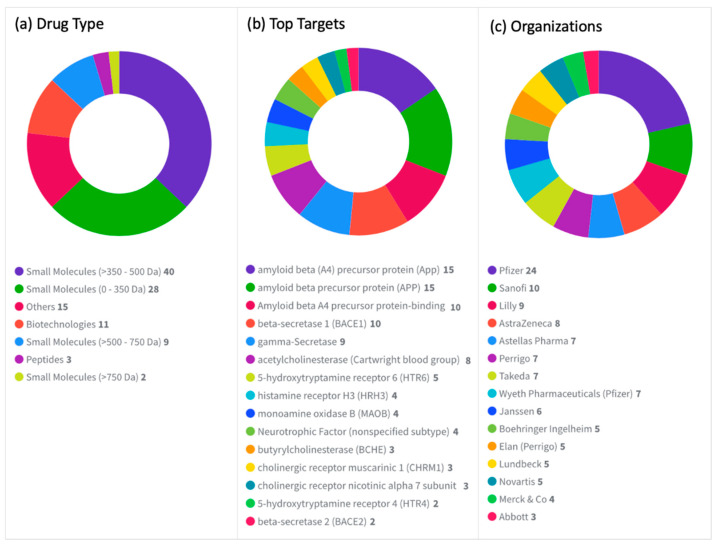
An overview of Alzheimer’s disease drugs that were either suspended, withdrawn, or discontinued by (**a**) drug type, (**b**) top targets, and (**c**) top organizations. Under active development, according to Cortellis Drug Discovery Intelligence (CDDI) [40] database as of 10 October 2022. Data source: Cortellis Drug Discovery Intelligence, 10 October 2022, https://www.cortellis.com/drugdiscovery/ © 2022 Clarivate. All rights reserved.

**Figure 6 diagnostics-12-02975-f006:**
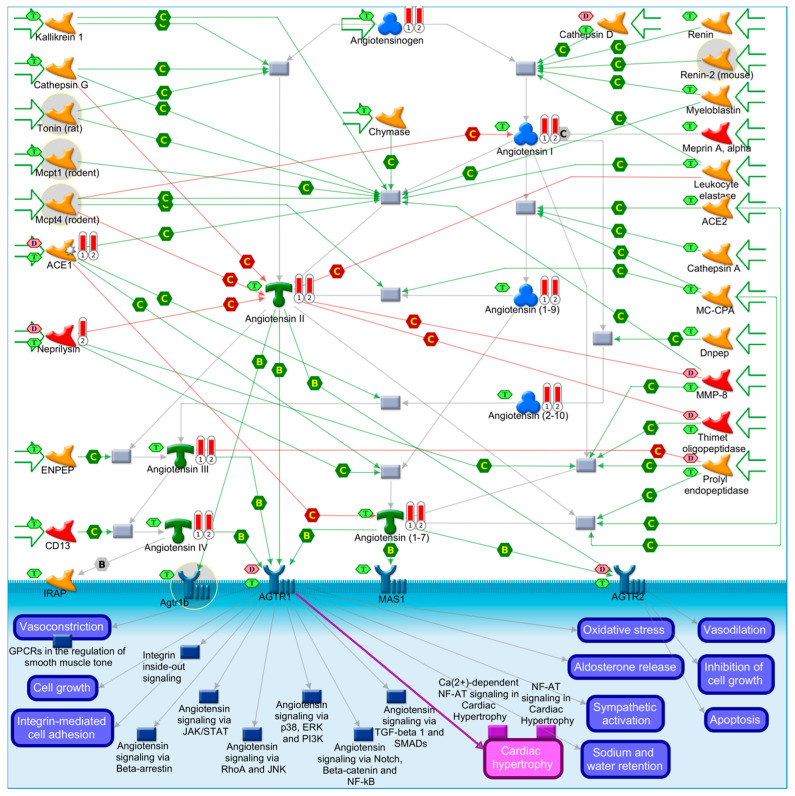
Top enriched pathway map for ‘common’ biomarkers of EOAD and LOAD. The pathway map representing protein-folding and maturation for the angiotensin system maturation. Connections between network objects on the map are referred to as links (or edges). A link identifies an interaction or a logical relation between two nodes. The type of interaction or relation is reflected by an appropriate symbol placed in the middle of the link. B = binding (i.e., physical interaction between molecules); C = cleavage of a protein at a specific site, yielding distinctive peptide fragments and carried out by enzymes or compounds; red arrows = inhibition; green arrows = activation; grey arrows = unspecified action; solid purple arrows = emergence in disease; light violet text box = normal process; pink text box = pathological processes; white text box with blue outline = notes; grey block = reaction; blue block = normal process; pink block = pathological process; starred network objects = groups or complex processes; red thermometers on pathway map = network object is a validated biomarker for AD, according to the Cortellis Drug Discovery Intelligence (CDDI) database [40]; thermometer 1 for EOAD, and thermometer 2 for LOAD; a pink hexagon with a capital D on the upper left side of the network object indicates an AD biomarker according to MetaCore^TM^; a green hexagon on the upper left side of the network object with a capital T indicates that the network object is expressed in the brain. Map generated using MetaCore^TM^ version 21.4. MetaCore^TM^, a Cortellis™ solution, 14 October 2022, © 2022 Clarivate. All rights reserved.

**Table 2 diagnostics-12-02975-t002:** Highest validity Alzheimer’s disease biomarkers listed according to their biomarker uses in Alzheimer’s disease.

	Biomarker Name	Population	Role	Highest Use Validity	Gene Symbol
1	Amyloid beta A4 protein	Mild Cognitive Impairment	Risk Factor	Recommended/Approved	APP
2	Amyloid beta A4 protein	All	Diagnosis	Recommended/Approved	APP
3	Amyloid beta A4 protein	Early Onset	Diagnosis	Recommended/Approved	APP
4	Apolipoprotein E	Mild Cognitive Impairment	Risk Factor	Recommended/Approved	APOE
5	beta-amyloid protein 42	Mild Cognitive Impairment	Risk Factor	Recommended/Approved	
6	beta-amyloid protein 42	All	Diagnosis	Recommended/Approved	
7	Glucose transporters and hexokinases	Mild Cognitive Impairment	Risk Factor	Recommended/Approved	
8	Glucose transporters and hexokinases	All	Diagnosis	Recommended/Approved	
9	Microtubule-associated protein tau	Mild Cognitive Impairment	Risk Factor	Recommended/Approved	MAPT
10	Microtubule-associated protein tau	All	Diagnosis	Recommended/Approved	MAPT
11	Presenilin-1	All	Diagnosis	Recommended/Approved	PSEN1
12	Presenilin-1	Early Onset	Diagnosis	Recommended/Approved	PSEN1
13	Presenilin-2	All	Diagnosis	Recommended/Approved	PSEN2
14	Presenilin-2	Early Onset	Diagnosis	Recommended/Approved	PSEN2

**Table 3 diagnostics-12-02975-t003:** Approved symptomatic pharmacological treatments for patients with AD.

Drug	Drug Targets	Managed Symptoms	Mechanism of Action	Disease Stage
Donepezil [270,271,272,273,274]	AChE	Improves cognition and behavior	Cholinesterase inhibitor; inhibition of various aspects of glutamate-induced excitotoxicity; the reduction of early expression of inflammatory cytokines; the induction of a neuroprotective isoform of AChE; the reduction of oxidative stress-induced effects	Mild to moderate AD
Rivastigmine [272,273,274,275]	AChE; BChE	Improves cognitive functions and daily life activities	Cholinesterase inhibitor; increases cholinergic function	Mild to moderate AD
Galantamine [272,273,274]	AChE; nicotinic ACh receptor	Improves behavioral symptoms, daily life activities, and cognitive functions	Cholinesterase inhibitor; binds to α-subunit of nicotinic ACh receptors and activates them	Mild to moderate AD
Memantine [274,276]	NMDA receptor	Improves learning and memory	NMDA receptor antagonist (prevents over-activation of glutaminergic system that is involved in neurotoxicity in AD patients)	Moderate to severe AD

AChE: acetylcholine esterase; AD: Alzheimer’s disease; BChE: butyrylcholinesterase; ACh: acetylcholine; and NMDA: *N*-methyl-D-aspartate.

**Table 4 diagnostics-12-02975-t004:** Most important drug classes that failed as anti-AD treatments in different stages of clinical trials.

Drug Category	Classification	Why Suggested	Why Failed
Monoclonal Antibodies (mABs)	Disease-modifying	These antibodies target the amyloid protein, and they predominate drug discovery efforts [154]. Amyloid has been considered a promising drug target since it is located outside the nerve cells, and it is toxic to the brain’s tissues [154].	The mABs have not succeeded in eradicating AD because cognitive impairment predisposing dementia does not associate with amyloid precipitation [154].
Gamma (γ-) Secretase Inhibitors	Disease-modifying	It was proposed that targeting γ-secretase might reduce amyloid production, particularly Aβ42 isoform [160,161,162,163]. Phase II trials showed a dose-dependent decrease in both Aβ isoforms (Aβ40 and Aβ42) without significant decrease in tau protein, though the magnetic resonance imaging (MRI) recorded a cerebral atrophy following such treatment [154,165]. Patients showed some improvement at the beginning of treatment.	No distinct response of improvement nor worsening could be traced after 3 months of treatment [154,162]. Side effects were reported with higher doses, such as skin rashes, nausea, and diarrhea, accompanied by higher rate of skin cancer [154,164]. Furthermore, the narrow therapeutic window impeded their proceeding to Phase III [154,165].
Tau Inhibitors	Disease-modifying	The tau protein appeared as a potential target for AD dementia since an irregular phosphorylation of tau results in neurofibrillary tangle formation [166,167,168]. Clinical studies reported that AD progress is related to tangle formation more than that of Aβ [156]. Initially, tau aggregation inhibitors (TAIs) showed better response.	After long-term treatment (approximately 15 months), TAIs failed in AD treatment. Moreover, 15% of patients showed minor improvement without any co-administered therapy [169].
Neurochemical Enhancers	Symptomatic	Idalopiridine that inhibits 5-hydroxytryptamine 6 (5-HT6) receptors and consequently enhances the release of acetylcholine in the brain, i.e., pro-cholinergic effector [182,183]. Encenicline incites cholinergic response through activating α-7 nicotinic acetylcholine receptors [185,186,187].	Further clinical studies declared that Idalopiridine does not show any promising effect in AD treatment [182,184]. Side effects of Encenicline were observed in Phase II trials at the maximum dose (2 mg) [185,186,187]. In addition, the Phase III trials, with doses of 2–3 mg, were terminated due to GI toxicity and eventually discontinued because there was no improvement in cognitive function [185,186,187].
Miscellaneous	Symptomatic	Dimebon is a histamine (H1) antagonist [188]. It affects α-adrenergic and serotonergic receptors, AMPA and NMDA glutamate receptors, and L-type voltage-gated calcium channels [189].	It exerted a better response in AD patients and one Phase II trial in Russia [189], but it failed in Phase III trials in Austria, Europe, New Zealand, and the US [189].

## Data Availability

Data supporting the reported results can be requested by contacting the corresponding author directly.

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
