# Peer review of "A Review of the Recent Advances in Alzheimer’s Disease Research and the Utilization of Network Biology Approaches for Prioritizing Diagnostics and Therapeutics"

_diagnostics, 2022, doi:10.3390/diagnostics12122975_

Round 1

Reviewer 1 Report

This is a nice and comprehensive review on Alzheimer’s disease  providing a better understanding of disease pathogenesis hypotheses, including the role of genetic and epigenetic factors in disease development and progression.

The manuscript is well written, guiding the reader through the complex in the molecular mechanisms underlying the pathophysiology of Alzheimer’s disease. Authors made a big effort to make a complete literature revision, giving the correct biological context, and to create the conceptual links between different studies, including the most recent advances in the search of effective disease biomarkers, drug targets. The authors also suggest the application of machine learning and artificial intelligence to multi-omic Alzheimer's disease data extraction to facilitate drug and biomarker discovery efforts such as to hypothesize the possibility of personalized treatments in the future.

Author Response

We thank the reviewer for the nice feedback. 

Reviewer 2 Report

In the present article, the authors have elaborated the pathophysiology of AD and how the ML and AI can be included to diagnose and possible management of this neurodegenerative disease. 
The manuscript is well organised and can be accepted after following minor revisions. 
In figure 1, caption should be elaborated to explain the representation given in the figure. 
Rile if different genes, APOE, SORL, ABCA7 etc in AD should be represented diagrammatically for better understanding of readers. 
oxidative stress also plays a important role in Pathophysiolology of AD however it is not elaborated in manuscript. The authors are suggested to also focus on it. 
Conclusion and discussion should be enhanced for better outcome of the review process. 

Author Response

Response: We thank the reviewer for the helpful feedback.

1) In figure 1, caption should be elaborated to explain the representation given in the figure. 

Response: We elaborated on figure captions for figure 1 (and also figure 6).

Figure 1. Alzheimer’s disease (AD) pathway map highlighting validated diagnostic/prognostic AD biomarkers in addition to genes/gene products involved in the Tau pathology pathway. Con-nections between network objects on the map are referred to as links (or edges). A link identifies an interaction or a logical relation between two nodes. The type of interaction or relation is re-flected by an appropriate symbol placed in the middle of the link. B = binding (i.e., physical inter-action between molecules); C = cleavage; CM = covalent modification; +P = phosphorylation; -P = dephosphorylation; Tn = transport; TR = transcription regulation; red arrows = inhibition; green arrows = activation; grey arrows = unspecified action; solid purple arrows = emergence in dis-ease; dashed purple arrows = enhancement in disease; arrows with purple x = disruption in dis-ease; light violet text box = normal process; pink text box = pathological processes; white text box with blue outline = notes; grey block = reaction; blue block = normal process; pink block = patho-logical process; solid purple hexagon = compound; CS = complex subunit; GR = group relation; starred network objects = groups or complex processes; thermometers on pathway map = network object is also a validated diagnostic or prognostic disease biomarker from the Cortellis Drug Discovery Intelligence (CDDI) database [41]; thermometer 1 is for late onset AD (LOAD); ther-mometer 2 is for early-onset AD (EOAD); a pink hexagon with a capital D on the upper left side of the network object indicates the presence of substantial experimental evidence for involvement in Alzheimer's disease; a blue hexagon on the upper left side of the network object with a capital R indicates that the network object (i.e., gene/gene product) is a drug target (not necessarily for AD). Map generated using MetaCoreTM version 21.4.

Figure 6. Top enriched pathway map for ‘common’ biomarkers of EOAD and LOAD. The path-way map representing protein folding and maturation for the angiotensin system maturation. Connections between network objects on the map are referred to as links (or edges). A link identi-fies an interaction or a logical relation between two nodes. The type of interaction or relation is reflected by an appropriate symbol placed in the middle of the link. B = binding (i.e., physical in-teraction between molecules); C = cleavage of a protein at a specific site yielding distinctive pep-tide fragments, and carried out by enzymes or compounds; red arrows = inhibition; green arrows = activation; grey arrows = unspecified action; solid purple arrows = emergence in disease; light violet text box = normal process; pink text box = pathological processes; white text box with blue outline = notes; grey block = reaction; blue block = normal process; pink block = pathological pro-cess; starred network objects = groups or complex processes; red thermometers on pathway map = network object is a validated biomarker for AD according to the Cortellis Drug Discovery Intel-ligence (CDDI) database[41]; thermometer 1 for EOAD, and thermometer 2 for LOAD; a pink hexagon with a capital D on the upper left side of the network object indicates the presence of ex-perimental evidence for involvement in AD; a green hexagon on the upper left side of the network object with a capital T indicates that the network object is expressed in the brain. Map generated using MetaCoreTM version 21.4.

2) Rile if different genes, APOE, SORL, ABCA7 etc in AD should be represented diagrammatically for better understanding of readers. 

Response: Since we already have 6 figures (and most of them are complex) we think that adding the gene details for these genes would be excessive. However, we provided the gene structure for APP as an example and provided the web resources (Ensembl gene browser and NCBI gene browser). Interested researchers can check these resources for details about the other genes.

3) oxidative stress also plays a important role in Pathophysiolology of AD however it is not elaborated in manuscript. The authors are suggested to also focus on it. 

Response: We thank the reviewer for this valuable comment. Unintentionally, we missed discussing oxidative stress. We added a new section: 2.3. The Mitochondrial Cascade (Oxidative Stress) Hypothesis (p.7-8)

Neuronal mitochondria are considered the main organelles responsible for the neuronal oxidative stress via the generation of free radicals through its electron transport chain [109]. In the case of high levels of ATP and diminishing electron transport effect, superoxide would be formed from oxygen via mitochondrial respiration [110]. Superoxide would then be converted to hydrogen peroxide by superoxide dismutase and later to hydroxyl radicals and anions by the Fenton reaction [111]. These reactive oxygen species (ROS) would affect redox imbalance, cause neurotoxicity and genomic instability, tran-scription of pro-inflammatory genes, and cytokine release [112]. ROS would further damage and inactivate parts of mitochondrial electron transport chain leading to the formation of superoxide from the electron reduction of oxygen in a positive feedback cycle [110,112,113].

ROS damage mitochondrial DNA (mtDNA) that leads to neuronal functional im-pairments and damaged mitochondria due to this oxidative stress will not be degraded by mitophagy [109,112,114]. Usually, oxidative stress acts as a signal for mitophagy process to degrade damaged mitochondria as oxidative stress reduces mitochondrial membrane potential [109]. However, ROS alter parkin, which is a key mitophagy regulator, and inhibits its function causing the continual presence of mitochondria [114]. Collectively, ROS, DNA damage, and mitochondria contribute to the aging process [115,116]. Briefly, following DNA damage via ROS, kinases and PARP are activated leading to the decrease of NAD+ production, which is essential for metabolic pathways and ATP production. Thus, oxygen consumption and ATP production would be required to increase to meet high energy demand. Mitochondrial coupling would occur that increases membrane potential, increases free radical formation, and decrease mitophagy. And as discussed above, free radicals would further cause DNA damage. 

Denham Harman had proposed the free radical theory of aging, in which free radicals are involved in the changes associated with aging process [117]. This was later confirmed that free radicals are involved in the aging process and advanced age diseases, such as AD [116,118,119]. Furthermore, the effect of ROS on mitochondria supported the theory that relates mitochondria to the aging process and neurodegenerative diseases, such as AD [109]. This theory is mostly associated with the central nervous system since it consumes 20% of the body’s oxygen and is susceptible to oxidative stress [120]. Neurons would have high sensitivity to free radicals since they are non-dividing and post-mitotic cells and cannot be replaced in the events of damage leading to mitochondrial dysfunction with aging [119,121].

It is observed that mitochondrial dysfunction is prevalent in the aging process [109]. In addition, healthy aging results in reduced mitochondrial metabolism in terms of α subunit reduction of the mitochondrial F1 ATP synthase, thus interfering with ATP production [109]. Eventually, ATP production decreases, ROS production increase, and this will cause an increase in DNA, protein, and lipid oxidation [119,122,123]. Moreover, apart from mtDNA damage caused by mitochondrial dysfunction, nuclear DNA is also damaged leading to impairments in vesicular function, synaptic plasticity and mito-chondrial function [122].

Excess ROS and bioactive metals, such as copper, iron, zinc, and magnesium are present in AD brains that promote Aβ aggregations and NFT formation [124-127]. Moreover, the elevated levels of mtDNA oxidation are considered as one of the early markers of AD pathogenesis [119,123]. Also, late onset AD pathogenesis may be linked to age-associated mitochondrial decline [109]. The expression and processing of APP would be altered with age-associated mitochondrial decline leading to the production of Aβ oligomers that aggregates into plaques in AD [128,129]. These Aβ oligomers are asso-ciated with neuronal toxicities and ROS generation. Studies had shown that he hydro-phobic 25-35 region of Aβ leads to neuronal toxicity and generates ROS showing that Aβ itself is a source of oxidative stress [109,130,131]. Aβ42 is hydrophobic in nature, and it can reside within the neuronal membrane lipid bilayer and cause lipid peroxidation, iden-tified through 4-hydroxy-2-trans-nonenal (HNE) production that is bound to neuronal proteins [132]. Studies had shown also that the production of HNE, due to lipid peroxi-dation associated with the residing of hydrophobic Aβ protein in lipid bilayer, and HNE’s neuronal protein binding is linked to cell death [133-135]. This are related to neuro-degenerative diseases’ pathogenesis including AD. Furthermore, the oxidative stress triggered by Aβ is also likely to be as result of complexation with redox active metals, such as copper, zinc, and iron [109], which are highly present in AD brains [124-127], pro-moting Aβ aggregation into plaques [109]. Copper forms the most stable complex and can generate superoxide and hydrogen peroxide that contribute to AD pathogenesis [128,136].

4) Conclusion and discussion should be enhanced for better outcome of the review process. 

We enhanced different parts of the manuscript as well as the conclusion and abstract. All enhanced sections are highlighted in green in the modified manuscript file. And all additions are highlighted in yellow in the modified manuscript file.

Enhanced conclusion:

“AD is a multifactorial and polygenetic disease. Novel disease diagnostic biomarkers and disease-modifying treatments are required to halt or slow the onset and disease progression, decrease behavioral aberrations, and ameliorate cognition in AD patients. The recent advances in network biology approaches coupled with the advances in clinical trial design and protocols, in additional to the availability of powerful machine learning and artificial intelligence algorithms, hold the promise to identify novel diagnostic biomarkers, better drug targets, and effective disease-modifying drugs. Herein, we provide a comprehensive review on Alzheimer's disease highlighting the mainstream hypotheses explaining disease pathophysiology as well as current disease treatments and drug discovery projects. We also emphasize the recent scientific evidence implicating epigenetic mechanisms and the microbiome in AD pathogenesis and progression. We suggest that the application of Al and ML approaches in analyzing AD network biology derived from AD data including genetic, transcriptomic, epigenetic and metagenomic data would revolutionize our understanding of the disease pathways and will lead to the discovery of novel biomarkers and drug targets. Ultimately, these studies will increase our chances of identifying validated diagnostic biomarkers and effective disease-modifying cures. Hence, breakthrough discoveries in AD research are more likely to occur in the near future. This review provides a summary of the current hypotheses regarding AD pathogenesis in addition to most recent advances in the search of effective disease biomarkers, drug targets. This review also details AD drugs in various stages of development, and highlights technologies that are expected to accelerate AD drug and biomarker discoveries.”
